# Methanol-Enhanced Fe(III) Oleate-Catalyzed Aquathermolysis of Heavy Oil

**Rui Guo [1], Wei Fu [2], Le Qu [1], Yongfei Li [1], Weihua Yuan [1,3] and Gang Chen [1,4,***

[1] State Key Laboratory of Petroleum Pollution Control, Xi'an Shiyou University, Xi'an 710065, China
[2] Xinjiang Xinchun Petroleum Development Co., Ltd., SINOPEC, Dongying 257092, China
[3] Shaanxi Province Key Laboratory of Environmental Pollution Control and Reservoir Protection Technology of Oilfields, Xi'an Shiyou University, Xi'an 710065, China
[4] Xi'an Key Laboratory of Tight Oil (Shale Oil) Development, Xi'an Shiyou University, Xi'an 710065, China
[*] Correspondence: gangchen@xsyu.edu.cn

**Abstract:** Fe(III) oleate (Fe(III)L) has been used in heavy oil aquathermolysis as catalysts, but the effect of the hydrogen donor on this reaction has not been considered. In this paper, we introduce methanol as the hydrogen donor in the Fe(III)L-catalyzed aquathermolysis to investigate the promotion effect of methanol on the aquathermolysis. The results show that the addition of methanol can increase the viscosity reduction rate of aquathermolysis from 81.81% to 91.23%. The heavy oil samples before and after aquathermolysis were characterized by thermogravimetric (TGA), differential scanning calorimetry (DSC), elemental analysis (EA), and carbon number distribution to investigate the changes in physical and chemical properties and explore the mechanism of methanol as a hydrogen promoter. There was a significant decrease in asphaltene and resin in the oil sample subjected to the reaction after the addition of methanol; the wax precipitation point decreased from 38 °C to 31 °C; the S element content decreased by 1% and the C element content increased by 4%; the content of light saturated HC (less than C10) increased and the content of saturated HC with more than C10 decreased. It shows that the addition of methanol, which provides a large amount of active hydrogen, promotes the breakage of long-chain alkanes in heavy oil, the light component content increase, promotes the breakage of C–C and C–S bonds during the reaction, making the content of heteroatoms decrease, increases the viscosity reduction rate, and improves the fluidity of oil samples. The findings of this study can help for better understanding of the mechanism of methanol in aquathermolysis and facilitate the exploration and exploitation of heavy oil.

**Keywords:** heavy oil; aquathermolysis; viscosity reduction

## 1. Introduction

With the continuous development of human society, the demand for oil resources is increasing. The world is rich in oil resources, of which heavy oil resources account for about 90% of conventional oil resources [1–5]. However, heavy oil contains a large amount of resin and asphaltene, which leads to poor fluidity and difficulty in heavy oil recovery [6]. Improving the fluidity of heavy oil has become an important prerequisite for heavy oil development, and many research achievements and attempts have been made [7–10]. Aquathermolysis has attracted much attention as an emerging technology, especially as catalyzed aquathermolysis [11].

Oleic acid molecules have good compatibility with oil and high thermal stability, making it well-suited ligands in aquathermolysis catalyst preparation [12]. In aquathermolysis, with transition metal as the active center, it is easier to penetrate the recombinant component for the reaction [13]. Wen et al. used molybdenum oleate as a catalyst in aquathermolysis for heavy oil of the Liaohe oilfield, and the viscosity was reduced by more than 90% [14]. The results show that the catalytic effect of the molybdenum oleate catalyst is more effective than the inorganic catalyst. Feoktistov et al. used the oil-soluble complex of Fe as the

catalyst for aquathermolysis [15]. The results show that the content of high molecular weight components in the oil sample decreased significantly. Research indicates that oleic salts have a good effect on the viscosity reduction of heavy oil. Li et al. prepared Fe(III) oleate and applied it to the low-temperature aquathermolysis of Shengli crude oil. It is more active than Ni and Co, and the viscosity reduction rate of heavy oil is as high as 86.1% [16]. In terms of viscosity reduction by aquathermolysis, the Fe(III) oleate catalyst has been a mature catalyst and has been widely used. To further improve the viscosity reduction of heavy oil, the choice of hydrogen promoter is the top priority. A large number of free radicals generates during the aquathermolysis. Since the amount of active hydrogen in the reaction is not enough to allow all free radicals to react, a large number of free radicals re-polymerize after the reaction, resulting in the rebound of heavy oil viscosity [15]. If a substance that can provide a large amount of active hydrogen in the reaction is added to the reaction, the aquathermolysis is greatly promoted, and the viscosity of heavy oil is reduced to a greater extent. Research shows that water is a hydrogen promoter in the aquathermolysis of heavy oil [17], but its efficiency is limited. Although it can promote aquathermolysis to a certain extent, it is necessary to find a more effective hydrogen promoter. Zhao et al. used formamide as a hydrogen promoter to study the effect on aquathermolysis [18]. The data verifies that after adding hydrogen donor, the content of the light component increases, and the content of the heavy component decreases furtherly. Zhou et al. studied the effect of different alcohols as hydrogen donors on aquathermolysis, and the experiments show that the addition of alcohol is beneficial to decreasing the viscosity, among which methanol as hydrogen donor is the most suitable, and the decrease of viscosity reaches 87% [19]. It can be seen that the addition of the hydrogen promoter has a strong promoting effect on the aquathermolysis, and alcohol, as a hydrogen promoter, can provide a large amount of active hydrogen in the reaction, which is the most suitable hydrogen donor for the aquathermolysis, of which methanol is the most effective.

Oil-soluble aquathermolysis catalyst Fe (III) oleic acid (Fe(III)L) has become a well-established catalyst and has been widely used, but studies focusing on the mechanism of action of hydrogen cocatalyst in this system are still in the blank, so investigating the role of methanol as a hydrogen donor in this aquathermolysis is meaningful for applications. In this work, we optimized the factors affecting the aquathermolysis and employed a variety of characterization means to explore the effect of hydrogen donor addition on the rate of viscosity reduction of the aquathermolysis as well as to explore the mechanism of hydrogen donor action in the reaction, as shown in Figure 1

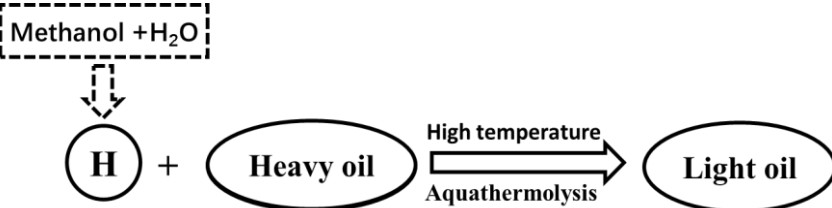

**Figure 1.** Reaction flow.

## 2. Materials and Methods

### 2.1. Materials

The reagents such as petroleum ether, toluene, sodium oleate, and FeCl$_3$ are of AR grade. The heavy oil used for evaluations was Baikouquan crude oil from Xinjiang Oilfield. Its main properties are shown in Table 1.

**Table 1.** The physical parameters of Heavy oil.

| Freezing Point/°C | Moisture Content/% | Asphaltene, % | Saturated HC, % | Aromatic HC, % | Resin, % |
|---|---|---|---|---|---|
| 5.3 | 11.41 | 0.02 | 48.50 | 21.00 | 19.07 |

### 2.2. Synthesis of the Catalyst

The synthesis of this Fe(III) oleate was shown in Figure 2. A mount of sodium oleate (1 mmol) was dissolved in ethanol, an ethanol solution of $FeCl_3$ (3 mmol) was added dropwise, and the mixture refluxed for 4 h with stirring. The mixture was cooled to room temperature, and the supernatant was obtained after centrifugation. Then the solvent was evaporated from the supernatant, and the remaining was iron oleate.

**Figure 2.** Preparation of Fe(III) oleate.

### 2.3. Catalyzed Aquathermolysis of Heavy Oil

The experiments were conducted at different reaction temperatures (200 °C to 300 °C), different water-to-oil mass ratios (0 to 0.6), and catalyst-to-oil mass ratios (0.05), which were introduced into the reactor. The mixture was cooled to room temperature after the reaction and poured into a measuring glass for transport properties and compositional tests.

### 2.4. Product Evaluation

The viscosity of the heavy oil before and after the reaction was determined using a rotary viscometer (Shanghai Changji Geological Instrument Co., Ltd., Shanghai, China), and the water in the oil sample was removed by centrifugation before testing. Elemental analysis (EA) of the oil samples before and after the reaction was performed using an elemental analyzer (German Element Analysis System Co., Ltd., Germany). The carbon number distribution of saturated hydrocarbons in oil samples before and after the reaction was analyzed by gas chromatography (Agilent Technologies Co., Ltd., Palo Alto, CA, USA) internal standard method (cyclohexane as solvent). The temperature was increased in three stages, from 50 °C to 80 °C at a rate of 4 °C/min and kept for 3 min, then increased to 120 °C at a rate of 15 °C/min and kept for 3 min, and finally increased to 240 °C at a rate of 12 °C/min and kept for 13 min, $N_2$ was used as the carrier gas (flow rate 229.22 mL/min). Thermogravimetric analysis (METTLER TOLEDO, Zurich, Switzerland) (TGA) was performed using a TGA-DSC thermal analyzer in the temperature range of 30–500 °C with a ramp rate of 10 °C/min. The measurements were performed using $SiO_2$ crucibles in an $N_2$ atmosphere (flow rate 20 mL/min). The analysis was performed using a differential scanning calorimeter (METTLER TOLEDO, Zurich, Switzerland) (DSC) in the temperature range from −30 to 80 °C with a ramp rate of 11 °C/min and a cooling rate of 8 °C/min. Measurements were performed in an $N_2$ atmosphere (flow rate 20 mL/min). The morphology of paraffin crystals in oil sample saturated hydrocarbons before and after the reaction was observed using BK-POL polarizing microscope (Chongqing Auto Optical Instrument Co., Ltd., Chongqing, China).

## 3. Results and Discussion

### 3.1. Effect of Reaction Temperature

The effect of temperature on the viscosity of oil samples is of primary importance, and the effects of different reaction temperatures on the viscosity of oil samples were compared, as shown in Figure 3. The data indicate that increasing the reaction temperature is beneficial to some extent for the reduction of oil sample viscosity. At 250 °C, the viscosity of the oil sample was reduced from 750,000 to 347,000 mPa·s, and the viscosity reduction effect was quite significant.

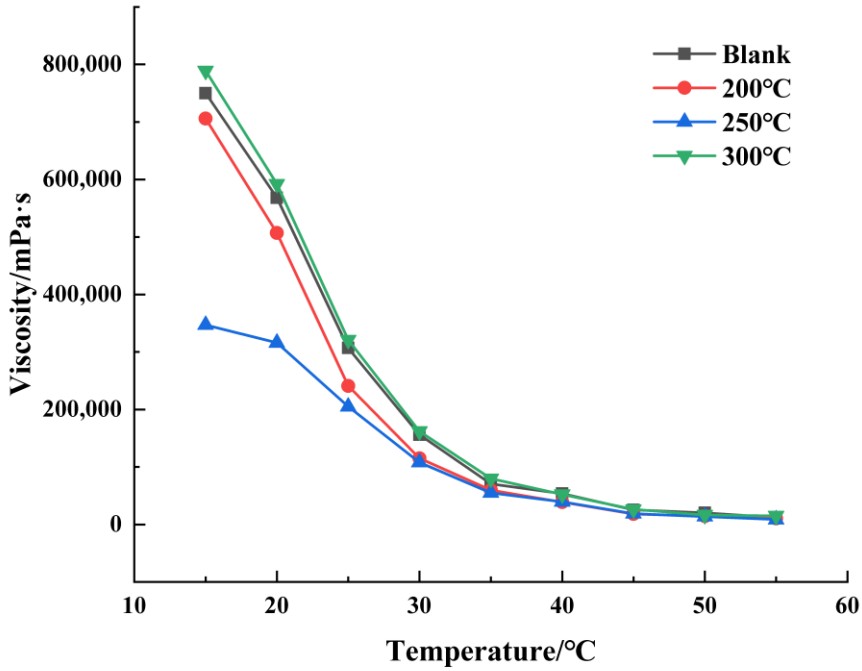

**Figure 3.** Effect of different reaction temperatures on oil sample viscosity.

### 3.2. Effect of Water

The effect on oil sample viscosity after aquathermolysis reaction at different mass ratios of water-to-oil is shown in Figure 4 [20]. The data show that after the aquathermolysis, the viscosity of the heavy oil decreased significantly. The addition of water can reduce the viscosity of the oil sample. When the mass ratio of water-to-oil reaches 0.3, the viscosity reduction effect is the most obvious, which could be used as the reference concentration.

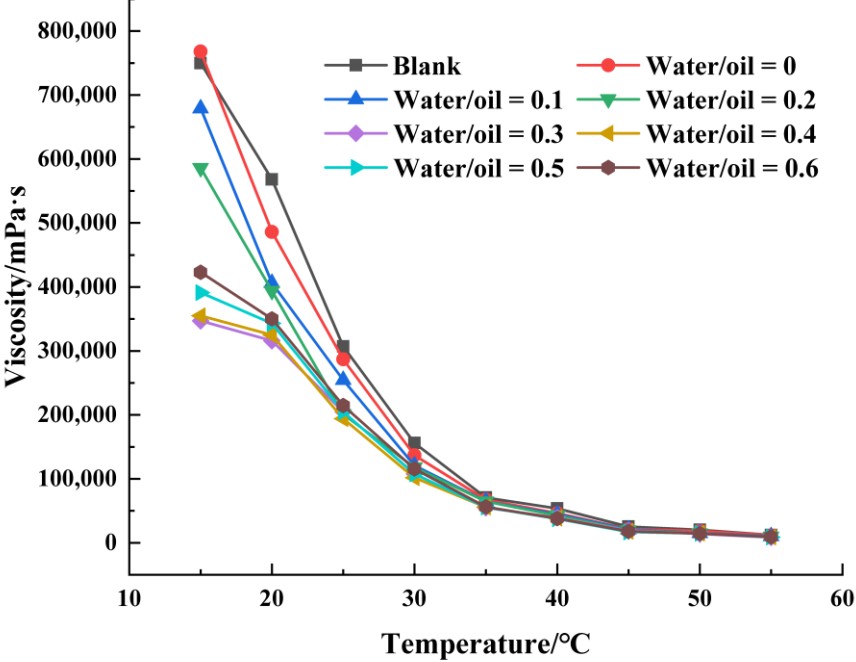

**Figure 4.** Effect of water on the viscosity of crude oil after aquathermolysis reaction.

### 3.3. Effect of Methanol as Hydrogen Donor

Methanol was added to the reactants containing water and oil with a mass ratio of 0.2. The changes in viscosity before and after the oil sample reaction are shown in Figure 5.

At 15 °C, the viscosity drops to 68,600 mpa·s, which is 80.23% lower than that of the oil sample with only water, and 32% lower than that of the oil sample with Fe(III)L, indicating that the catalyst and methanol have good synergy [21]. The addition of methanol can better promote the reduction of the viscosity of heavy oil.

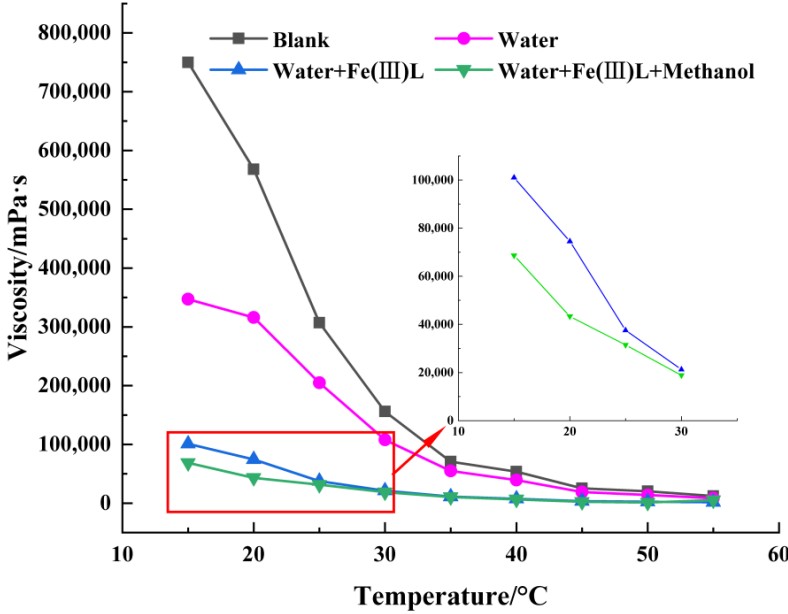

**Figure 5.** Effects of methanol on the viscosity of the oil sample.

### 3.4. Elemental Analysis and Carbon Number Distribution (EA)

At the element level, after adding hydrogen promoter, the carbon content increased from 78.4% to 86.0%, and the heteroatom decreased from 2.89% to 1.95%, as shown in Figure 6. The increase in carbon is due to the addition of catalyst and methanol, which may result in the decrease of N and S content. The heteroatom loss is due to the production of water-soluble compounds such as alcohols and sulfides [22–24]. It proves that the addition of methanol does facilitate the aquathermolysis to proceed, improving the catalytic aquathermolysis and removing some heteroatoms.

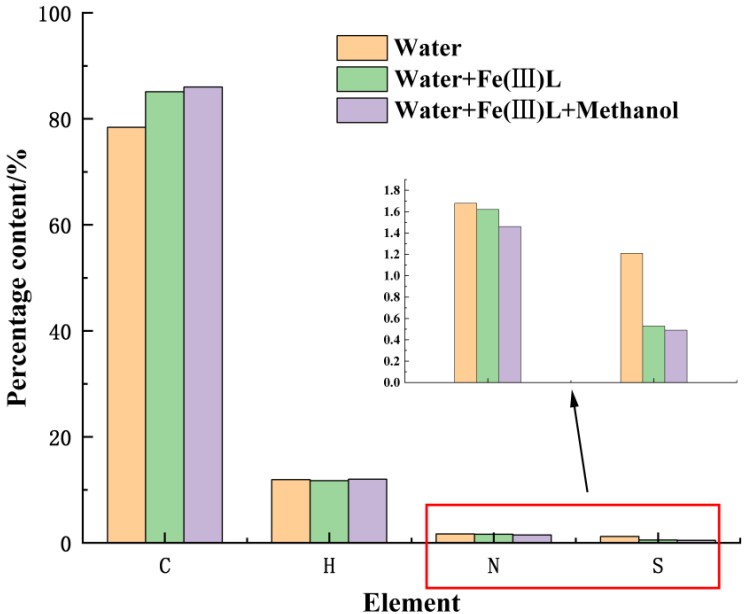

**Figure 6.** Changes in the heavy oil element content before and after the reaction.

The carbon number distributions of the saturated hydrocarbons before and after the reaction are shown in Table 2. The contents of components < C10 increase and those > C10 decrease in the presence of water, catalyst and methanol. The results show that the addition of methanol decreases the components above C15 while suppressing the polymerization.

**Table 2.** Carbon number distribution of heavy oil before and after the reaction.

| Carbon Number Distribution | Blank/% | Fe(III)L/% | Fe(III)L + Methanol/% |
|---|---|---|---|
| <C10 | 49.39 | 73.59 | 75.06 |
| C10~C15 | 38.21 | 19.91 | 22.12 |
| C15~C20 | 11.71 | 6.50 | 2.82 |
| >C20 | 0.69 | 0 | 0 |

*3.5. Thermogravimetric Analysis (TGA)*

TGA reveals some changes in the volatility of the components as a result of the reaction [25,26]. The results in Figure 7 show a similar pattern before and after the reaction. The weight loss of the oil samples before and after aquathermolysis at 50–200 °C increases from 31.3% to 33.5%, and the weight loss increased slightly, which is consistent with the production of a small amount of lighter molecules, and the weight loss at 350–500 °C decreases from 27.3% to 21.9%. The results show that the addition of methanol can transform more heavy components into light components and decrease the viscosity and pour point.

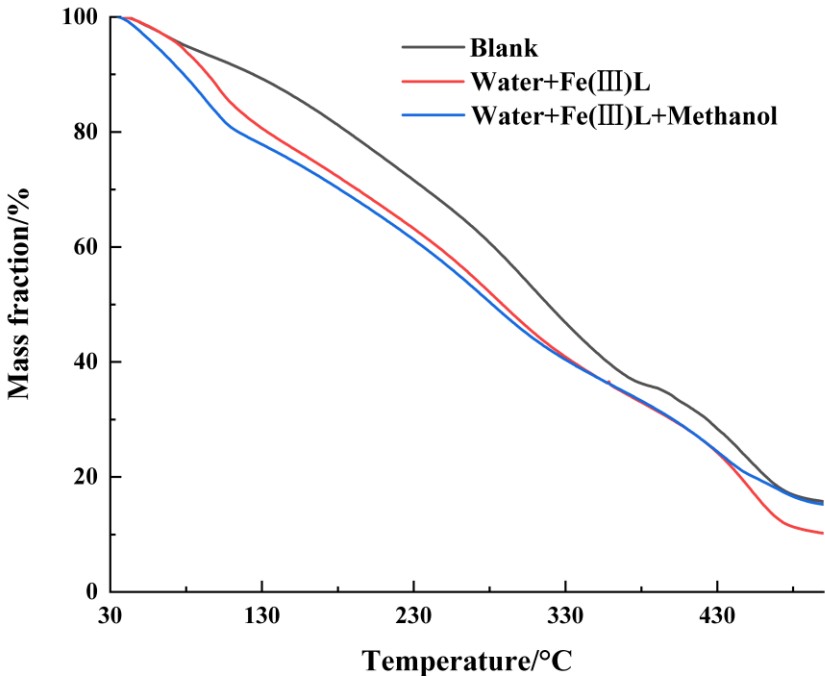

**Figure 7.** The TG curves of the crude oil before and after aquathermolysis.

*3.6. Differential Scanning Calorimetry Analysis (DSC)*

The wax content in heavy oil is an important factor determining the low-temperature fluidity of heavy oil, and we investigated the wax formation process of oil sample before and after aquathermolysis. The main component of wax content is high carbon number alkanes. In the cooling process of heavy oil, wax molecules with high carbon number firstly precipitated to accelerate heavy oil solidification. As shown in Figure 8, there was an obvious left shift after the addition of methanol compared with the DSC curve of the hydrothermal decomposition of the oil sample using only the catalyst. The wax evolution point decreased from 38 °C to 31 °C. It was shown that these high carbon number alkanes

are converted to low carbon number alkanes during the progress of aquathermolysis. At the same time, the content of saturated hydrocarbons in the oil samples increased after the reaction, and the content of soluble waxes also increased, thus slowing down wax crystallization precipitation and reducing the wax evolution point of the oil sample.

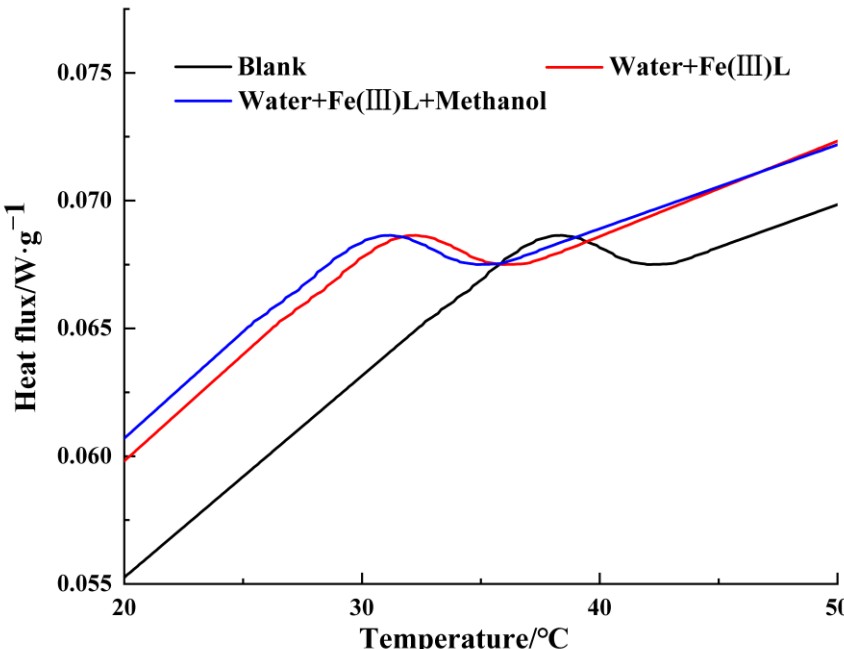

**Figure 8.** DSC of oil samples with and without Fe(III)L and methanol.

### 3.7. Paraffin Crystals

The morphology and structure of paraffin crystals have an extremely important influence on the rheology of heavy oil, and we investigated the changes in wax crystal morphology in the oil sample before and after the reaction, as shown in Figure 9. Paraffin crystals in crude oil are larger and more numerous, and the crystal morphology is fine and needle-shaped. After the addition of methanol, paraffin crystals significantly decreased in number, decreased in size, increased dispersity, and the distribution became more sparse [27], and the crystal morphology changed from needle-shaped to regular spheroid-like junction crystals with the weaker association, resulting in improved heavy oil fluidity.

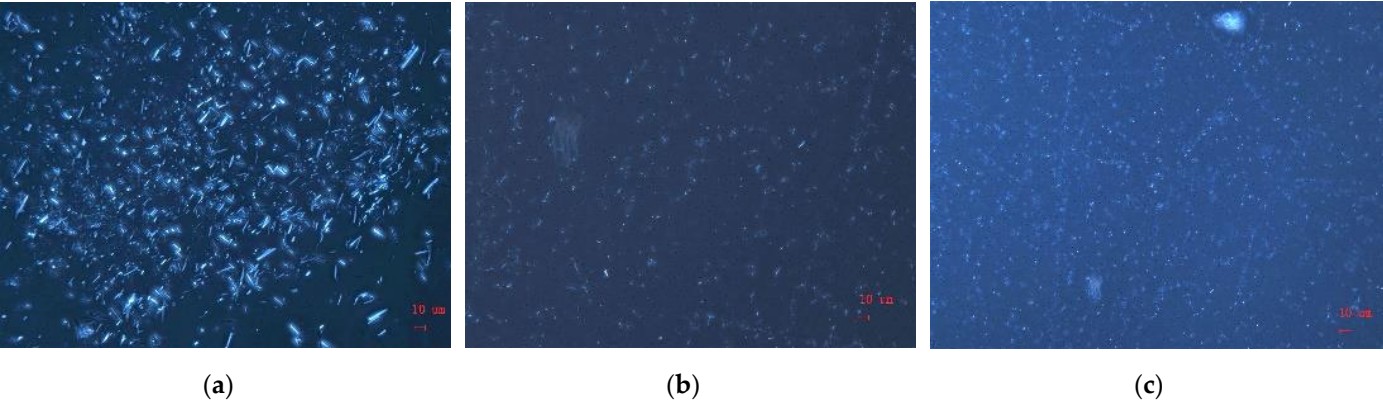

(**a**)  (**b**)  (**c**)

**Figure 9.** Photography of paraffin crystal in saturated HC of the heavy oil before and after aquathermolysis. (**a**) Blank; (**b**) Fe(III)L; (**c**) Fe(III)L + methanol.

### 3.8. Mechanism

Hydrogen donors can act on the asphaltene of heavy oil and improve the quality of heavy oil. Under the condition of high temperature and high pressure, the hydrogen donor can produce active hydrogen and react with active radical fragments to prevent the reverse combination of free radicals and the formation of coke [28,29]. The action mechanism is shown in Figure 10.

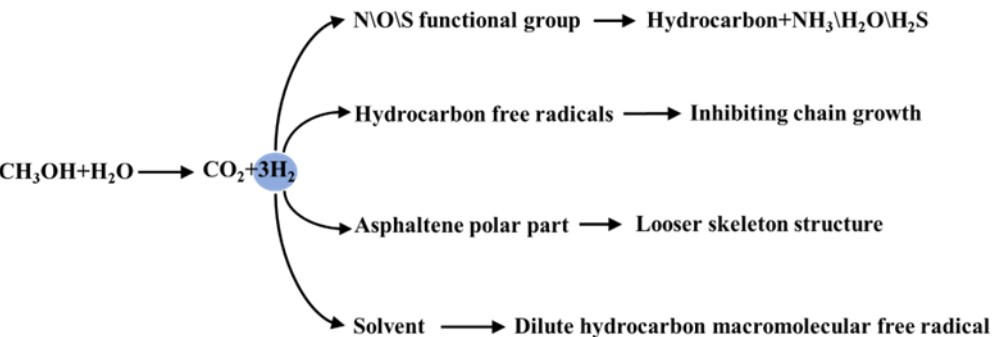

**Figure 10.** Reaction mechanism.

When methanol is added to the reaction, the reaction of methanol with water under high temperature and high pressure conditions will produce a large amount of active hydrogen, carbon monoxide, as well as a small amount of marsh gas, and the single-pass conversion can reach more than 99%. The reaction equation is shown in (1).

$$CH_3OH + H_2O \rightarrow C_2O + CO + H^\bullet \tag{1}$$

$$\text{Nitrogenous compound} + H^\bullet \rightarrow NH_3 \tag{2}$$

$$\text{Oxygenated compounds} + H^\bullet \rightarrow H_2O \tag{3}$$

$$\text{Sulfur compounds} + H^\bullet \rightarrow H_2S \tag{4}$$

$$R^\bullet + H^\bullet \rightarrow RH \tag{5}$$

Active hydrogen interacts with the polar part of the recombinant component in heavy oil, reduces the interaction within and between the high polar molecules, and loses the intermolecular action of resin and asphaltene, which is conducive to aquathermolysis. The presence of active hydrogen inhibits the condensation reaction, delays the aggregation and coking reaction of asphaltene [30], and improves the conversion rate of aquathermolysis. Meanwhile, the generated $CO_2$ dilutes the heavy oil and reduces the viscosity.

Moreover, active hydrogen converts the high polar components containing N, O, and S into low polar components and $NH_3$, $H_2O$, and $H_2S$ [31], as shown in (2)–(4), reducing the hydrogen bond between resin and asphaltene molecules [32], so as to reduce the viscosity of heavy oil. The presence of active hydrogen will annihilate some free radicals produced in the aquathermolysis, reduce the number of free radicals in the reaction (as shown in (5)), dilute the concentration of macromolecular free radicals, inhibit the condensation reaction and coking reaction, reduce the asphaltene component in heavy oil, and reduce the viscosity of oil sample as a result.

### 4. Summary and Conclusions

In summary, in this paper, the promoting effect of methanol in Fe(III) oleate-catalyzed aquathermolysis was investigated to reveal the mechanism of methanol as a hydrogen donating agent. When methanol was added, the viscosity reduction rate after aquathermolysis increased from 81.81% to 91.23%. When methanol is added to the system, more active hydrogen will be provided during the reaction, more radicals generated during the reaction are captured, and more light components are produced, promoting the breakage of long carbon chains, removing some heteroatoms, disrupting the interaction between asphaltenes

and gums, promoting the conversion of asphaltenes to gums, improving the fluidity and quality of crude oil, and adding hydrogen donor methanol, which can effectively improve the aquathermolysis efficiency. However, the reaction was performed at 250 °C and the reaction temperature was still high, which needs to be optimized in subsequent studies.

**Author Contributions:** R.G.: data curation, investigation and writing—original draft; W.F.: validation, conceptualization, supervision; W.Y.: data curation, investigation, formal analysis; Y.L.: investigation, methodology, data curation; L.Q.: project administration, funding acquisition; G.C.: writing—review and editing, validation, and supervision. All authors have read and agreed to the published version of the manuscript.

**Funding:** This research was funded by National Science Foundation of China, grant number 50874092.

**Data Availability Statement:** Not applicable.

**Acknowledgments:** The authors also thank the support of The Youth Innovation Team of Shaanxi University and the work of Modern Analysis and Testing Center of Xi'an Shiyou University.

**Conflicts of Interest:** The authors declare no conflict of interest.

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
