# Peer review of "Methanol-Enhanced Fe(III) Oleate-Catalyzed Aquathermolysis of Heavy Oil"

_processes, doi:10.3390/pr10101956_

Round 1

Reviewer 1 Report

Dear Authors

I have reviewed your paper. Overall the paper is well organized. I have several comments to polish manuscript:

1.     The abstract should highlight the key findings of this study.

2.     It is suggested to discuss more about the findings of this study at the end of abstract.

3.     It is recommended to mention about the applications of this study at the end of abstract:
The findings of this study can help for better understanding of …

4.     It is suggested to add a figure in Section 1, which presents the general sketch of the problem under study.

5.     The introduction may contain the background information, motivation for the study, contributions, and the paper organization. A separate section for the review of literature is recommended to be expounded.

6.     The research gap should highlight in introduction.

7.     The author contributions need to highlight in the paper. The main different of this paper and previous studies need to highlight for readers.

8.     What are the advantages and disadvantages of this study? I recommend the authors to highlight this topic.

9.     What are the limitations of this study? I recommend the authors to highlight this topic.

10.   The quality of all the figures should be improved.

11.  The title for last section should be changed to "Summary and Conclusions".

12.  The conclusion should improve to highlight the key finding of this paper.

13.  It is suggested to add a nomenclature (including alphabetic letters, Greek letters, subscripts, and superscripts).

Author Response

  1. The abstract should highlight the key findings of this study.

Thank you very much for your suggestions, I have made corresponding revisions in the text, and marked the yellow.

  1. It is suggested to discuss more about the findings of this study at the end of abstract.

Thank you very much for your suggestions, I have made corresponding revisions in the text, and marked the yellow.

  1. It is recommended to mention about the applications of this study at the end of abstract:

The findings of this study can help for better understanding of …

Thank you very much for your suggestions, I have made corresponding revisions in the text, and marked the yellow.

  1. It is suggested to add a figure in Section 1, which presents the general sketch of the problem under study.

Thank you very much for your suggestions, I have made corresponding revisions in the text, and marked the yellow.

  1. The introduction may contain the background information, motivation for the study, contributions, and the paper organization. A separate section for the review of literature is recommended to be expounded.

Thank you very much for your suggestions, I have made corresponding revisions in the text, and marked the yellow.

  1. The research gap should highlight in introduction.

Thank you very much for your suggestions, I have made corresponding revisions in the text, and marked the yellow.

  1. The author contributions need to highlight in the paper. The main different of this paper and previous studies need to highlight for readers.

Thank you very much for your suggestions, I have made corresponding revisions in the text, and marked the yellow.

  1. What are the advantages and disadvantages of this study? I recommend the authors to highlight this topic.

Thank you very much for your suggestions, I have made corresponding revisions in the text, and marked the yellow.

  1. What are the limitations of this study? I recommend the authors to highlight this topic.

Thank you very much for your suggestions, I have made corresponding revisions in the text, and marked the yellow.

  1. The quality of all the figures should be improved.

Thank you very much for your suggestions, I have made corresponding revisions in the text, and marked the yellow.

  1. The title for last section should be changed to "Summary and Conclusions".

Thank you very much for your suggestions, I have made corresponding revisions in the text, and marked the yellow.

  1. The conclusion should improve to highlight the key finding of this paper.

Thank you very much for your suggestions, I have made corresponding revisions in the text, and marked the yellow.

  1. It is suggested to add a nomenclature (including alphabetic letters, Greek letters, subscripts, and superscripts).

Thank you very much for your suggestions, I have made corresponding revisions in the text, and marked the yellow.

Reviewer 2 Report

When methanol vapor is passed over heated zinc, carbon monoxide, hydrogen and small amounts of swamp gas (Jahn) are obtained.

The article obtained interesting data on the aquathermolysis of heavy oil, which contains few resins and asphaltenes, but a lot of paraffins. Destruction of paraffins during aquathermolysis has been described little.

The results of determining the Carbon number distribution are given. However, there is no description of the methodology for conducting the analysis of saturated hydrocarbons.

Under what conditions does the reaction of the interaction of methanol with water proceed. What degree of conversion is expected for such a reaction to occur at 250–300°C?

Author Response

When methanol vapor is passed over heated zinc, carbon monoxide, hydrogen and small amounts of swamp gas (Jahn) are obtained.

Yes, methanol steam produces carbon monoxide, hydrogen, and marsh gases at high temperatures, which have been stated in the text and marked yellow.

The article obtained interesting data on the aquathermolysis of heavy oil, which contains few resins and asphaltenes, but a lot of paraffins. Destruction of paraffins during aquathermolysis has been described little.

Thank you very much for your suggestions, I have made corresponding revisions in the text, and marked the yellow.

The results of determining the Carbon number distribution are given. However, there is no description of the methodology for conducting the analysis of saturated hydrocarbons.

Thank you very much for your suggestions, I have made corresponding revisions in the text, and marked the yellow.

Under what conditions does the reaction of the interaction of methanol with water proceed. What degree of conversion is expected for such a reaction to occur at 250–300°C?

The reaction of methanol interacting with water is carried out at high temperature and pressure, generally at 220 ℃-300 ℃. The expected conversion for such a reaction to occur at 250 ℃-300 ℃ is 99%

Round 2

Reviewer 1 Report

The comments have been addressed